# Bilateral Serous Retinal Detachment as a Complication of HELLP Syndrome

**DOI:** 10.3390/diagnostics13091548

**Published:** 2023-04-26

**Authors:** Cosmin Adrian Teodoru, Corina Tudor, Maria-Emilia Cerghedean-Florea, Horațiu Dura, Ciprian Tănăsescu, Mihai Dan Roman, Adrian Hașegan, Mihnea Munteanu, Carmen Popa, Mihaela Laura Vică, Horea Vladi Matei, Horia Stanca

**Affiliations:** 1Faculty of Medicine, “Lucian Blaga” University of Sibiu, 550024 Sibiu, Romania; 2Department of Ophthalmology, Sibiu County Emergency Clinical Hospital, 550245 Sibiu, Romania; 3Department of Ophthalmology, “Victor Babes” University of Medicine and Pharmacy, 300041 Timisoara, Romania; 4Department of Radiology and Medical Imaging, Sibiu County Emergency Clinical Hospital, 550245 Sibiu, Romania; 5Department of Cellular and Molecular Biology, “Iuliu Haţieganu” University of Medicine and Pharmacy, 400012 Cluj-Napoca, Romania; 6Institute of Legal Medicine, 400006 Cluj-Napoca, Romania; 7Department of Ophthalmology, “Carol Davila” University of Medicine and Pharmacy, 050474 Bucharest, Romania

**Keywords:** HELLP syndrome, retinal detachment, pre-eclampsia

## Abstract

HELLP syndrome is a pregnancy complication, putting at risk the life of mother and child, characterized by high blood pressure, elevated liver enzymes and low platelets. Serous retinal detachment is a rare complication of pregnancy and may be associated with HELLP syndrome. One of the most common symptoms is a decrease in visual acuity. A rare case of bilateral exudative retinal detachment associated with HELLP syndrome is described in a 38-year-old woman a few hours after delivery. Optical coherence tomography (OCT) showed an amount of subretinal fluid and macular edema. Use of systemic corticosteroids and careful management of blood pressure led to early resolution of subretinal fluid and a good recovery of vision. Her final best corrected visual acuity was 1 (decimal notation) in both eyes at 2 weeks after delivery. These types of cases are rarely reported and highlight the importance of increasing awareness of this pathology among ophthalmologists.

HELLP syndrome is a disorder that can occur during pregnancy. It is characterized by high blood pressure, hemolysis, elevated liver enzymes and low platelets [1]. This syndrome is rare (about 10% of pre-eclampsia and eclampsia cases), but has a high maternal (3%) and infant (24%) mortality rate. Early therapy of the syndrome and delivery of the fetus is recommended in these cases to avoid visual sequelae [2].

On the first postpartum day, the patient presented a new-onset headache behind the left orbit associated with decreased visual acuity in the left eye. Her ophthalmic history was unremarkable. On the ophthalmological examination, best corrected visual acuity (BCVA) was 0.8 (decimal notation) in the right eye and counting fingers in the left eye. Anterior segment examination was unremarkable for both eyes, and the intraocular pressure was normal. There was no afferent pupillary defect. Ophthalmoscopic examination revealed bilateral exudative retinal detachment (RD) more extended in the left eye (Figure 1B) than in the right eye (Figure 1A). We noticed a clear vitreous without any cells, indicating no inflammatory condition, and a few microhemorrhages were observed in both eyes in the retinal periphery. The optic nerve was unremarkable. Optical coherence tomography (OCT) confirmed ophthalmoscopic findings by showing the serous RD involving the macula (LE), with the central macular thickness being 273 μm in the right eye and 709 μm in the left eye (Figure 2A,B).

Computed tomography of the head (without contrast) was performed and showed no acute intracranial abnormality. Head magnetic resonance imaging (MRI) without contrast was also performed, showing a hyperintense signal in the left eye (Figure 3).

Combined steroid therapy (dexamethasone phosphate 4 mg/mL once a day), pentoxifylline (100 mg/5 mL once a day) and antihypertensive treatment (1000 mg/day methyldopa, Amlodipine 5 mg/day and Metoprolol 5 mg/once a day) was introduced. On follow-up examination two days later, the patient had rapid improvement in her visual acuity to 0.9 (decimal notation) and 0.6 in the right and left eyes, respectively. On the 14th day postpartum, her BCVA had improved to 1.0 in both eyes. Fundus examination and macular OCT showed complete resorption of the fluid (Figure 4).

It is important to highlight that pregnancy affects all body systems, including the visual system. Metabolic, vascular or hemodynamic changes in ocular structures have been reported [3]. In other cases, during pregnancy, some preexisting ocular diseases may be exacerbated [4]. However, retinal vascular involvement in young people is quite rare, and therefore other diseases such as hematological diseases should be excluded [5,6].

Hypertension can lead to ischemic changes caused by a disturbance of the external blood–retinal barrier. Cases of hypertensive chorioretinopathy with serous retinal detachment have previously been recorded and are associated with malignant hypertension, pre-eclampsia or HELLP syndrome during pregnancy. These changes can be explained by an increase in endogenous vasoconstrictor agents leading to vasoconstriction and fibrinoid necrosis of choroidal vessels. As a result, this leads to ischemia of the overlying retinal pigment epithelium (Elschnig spots) causing leakage into the subretinal space (serous retinal detachment) (Figure 5) [7,8].

Exudative retinal detachment is an unusual complication of hypertension during pregnancy (may occur in 0.2–2% of patients with severe pre-eclampsia and 0.9% of patients with HELLP syndrome) [9]. The pathogenesis of exudative retinal detachment in eclampsia is not well known. There are several theories such as damage to the choriocapillaris by ischemia or thrombosis due to disseminated intravascular coagulation [1]. The development of microthrombi in HELLP syndrome can cause blockages and poor blood perfusion of the choriocapillaris [10]. Although it may appear at any time during pregnancy, this condition most often occurs before or shortly after birth, as in our case [11,12]. Clinically, the most common symptom is a sudden decrease in vision. Other eye complaints such as diplopia, photopsia or visual field defects have been reported [4].

There are no clear recommendations guiding the management of exudative retinal detachment in these cases. In patients who develop this condition before delivery, it is advisable to terminate the pregnancy by cesarean section as soon as possible [13]. Although the use of corticosteroid therapy in HELLP syndrome is controversial, its benefit has been proven. Corticosteroid administration in patients with HELLP is a cost-effective drug that can be administered through different routes, improving platelet count and serum LDH levels, in addition to reducing hospital stays and blood transfusion rates [14]. The use of corticosteroids and meticulous management of blood pressure, in our case, led to fluid resorption and symptom improvement.

As regards visual prognosis, in most cases, the detachment resolves with a return to normal visual function, without any long-term sequelae within the first few weeks postpartum. Surgery intervention is usually not necessary, and improvement occurs with good clinical management including blood pressure control [15]. In our case, the improvement in visual acuity was observed a few days after starting treatment. However, proper follow-up is mandatory for any pregnant woman with visual difficulties.

## Figures and Tables

**Figure 1 diagnostics-13-01548-f001:**
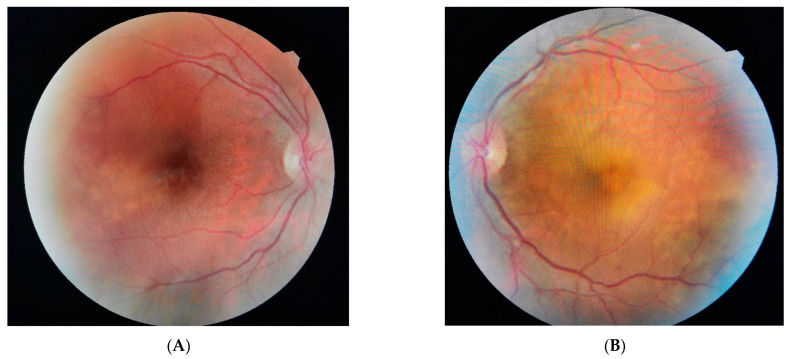
A 38-year-old primiparous woman presented to the emergency department at 35 weeks of gestation with headache and high blood pressure at 230/130 mmHg, heart rate at 97 beats per minute (bpm). Blood samples were drawn, showing the following results: increased inflammatory tests, increased coagulation tests, mild thrombocytopenia, elevated liver function tests and no proteinuria (Table 1). The patient was diagnosed with HELLP syndrome, and an emergency cesarean section was performed. She delivered a baby boy with a birth weight of 2020 g with a good Apgar score (9/5 min). (**A**) Right Eye. (**B**) Left eye.

**Figure 2 diagnostics-13-01548-f002:**
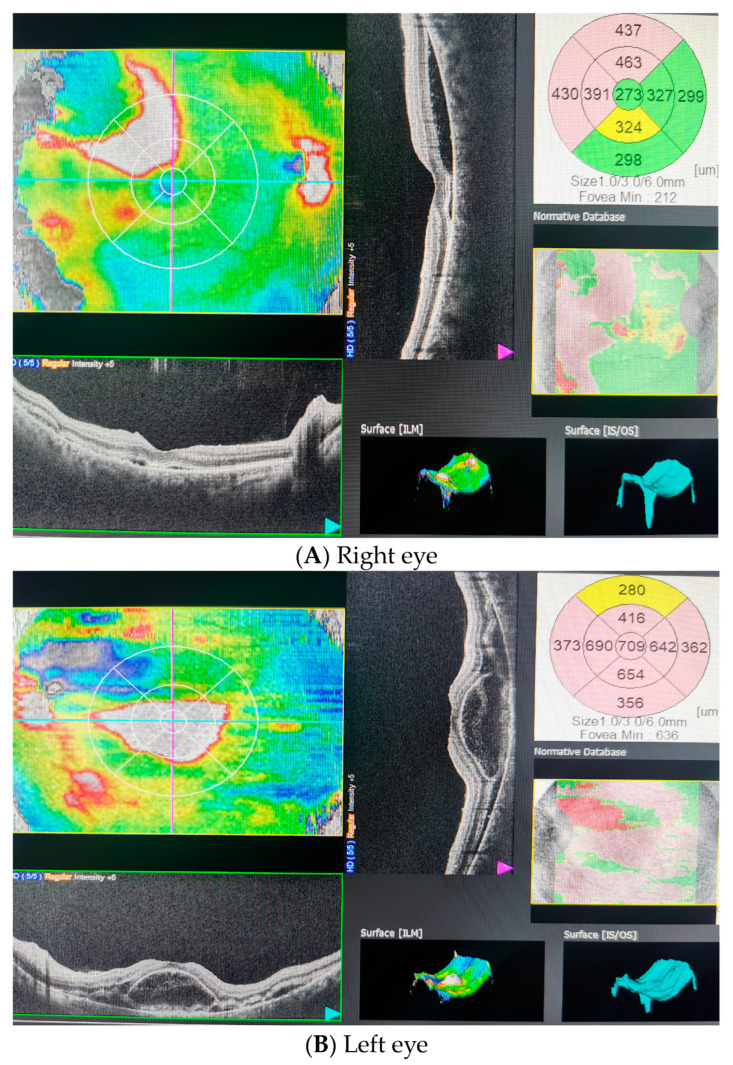
Optical coherence tomography (OCT) on the first day postpartum. Initial examination showed (**A**) subretinal fluid in the right eye and (**B**) intraretinal and subretinal fluid, retinal pigment epithelial detachment and bacillary layer detachment in the left eye.

**Figure 3 diagnostics-13-01548-f003:**
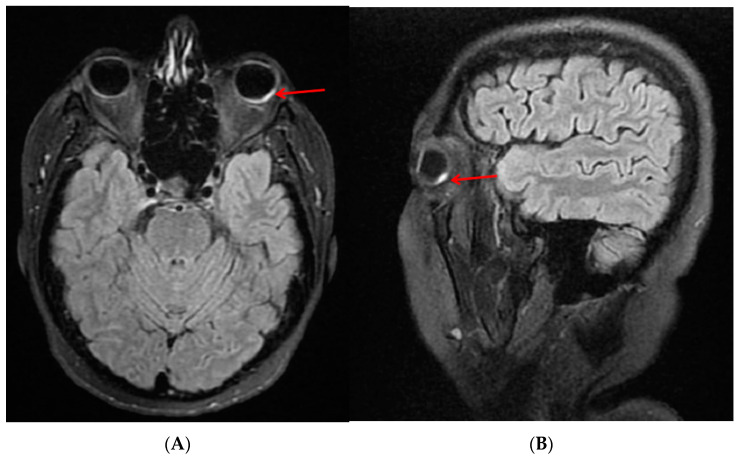
Head MRI without contrast (axial (**A**) and sagittal (**B**)) section showing parietal hyperintense FLAIR signal in the left eyeball, localized in the left superior posterolateral wall and right posteroinferior wall (red arrow).

**Figure 4 diagnostics-13-01548-f004:**
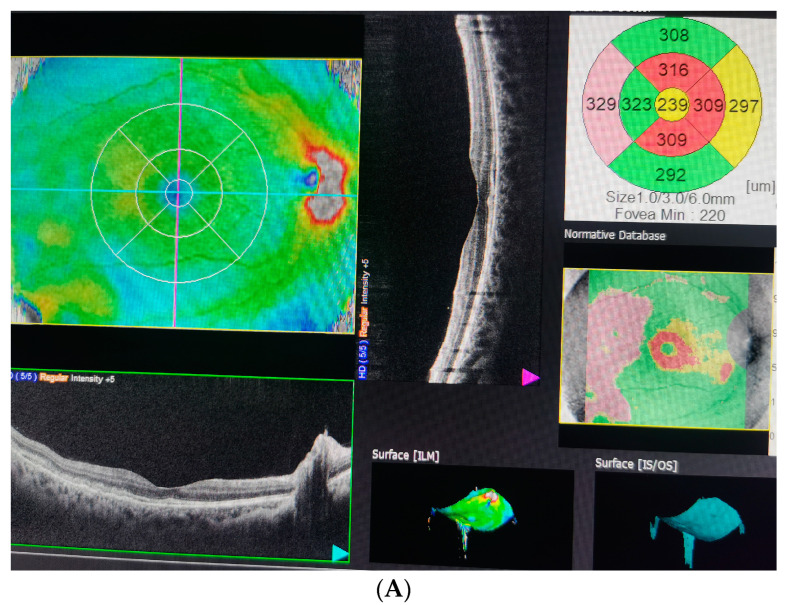
Optical coherence tomography (OCT) on the 14th day postpartum shows complete resorption of the fluid with an improvement in visual acuity in both eyes. (**A**) Right eye. (**B**) Left eye.

**Figure 5 diagnostics-13-01548-f005:**
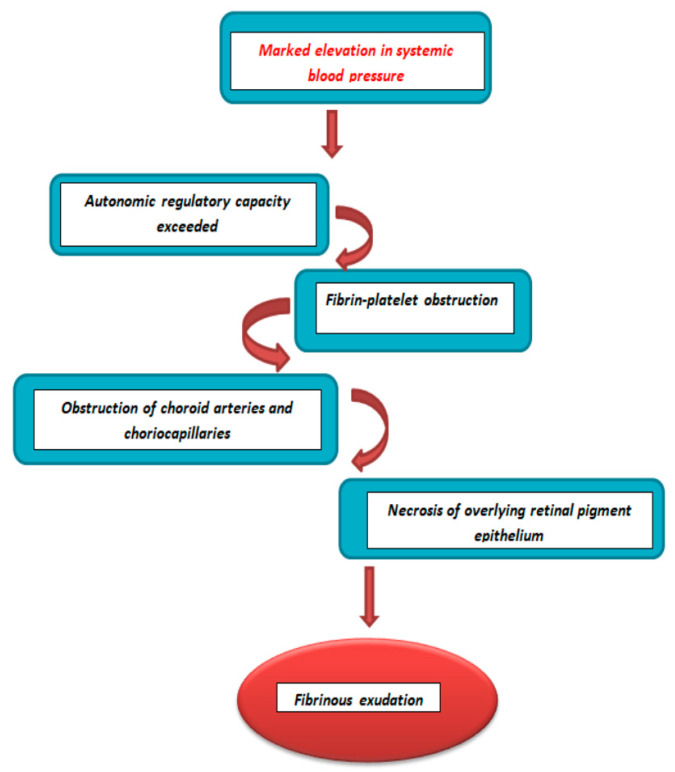
Choroidopathy hypertensive pathogenesis.

**Table 1 diagnostics-13-01548-t001:** Laboratory values for the patient.

Laboratory	Results
PCR (0–5 mg/dL)	102 mg/dL
Fibrinogen (179–420 mg/dL)	706 mg/dL
Procalcitonin (0–0.05 ng/mL)	2.74 ng/mL
D-dimers (45–499 ng/dL)	9560 ng/dL
Fibrinogen (170–420 mg/dL)	705 mg/dL
International Normalized Ratio (INR) (0.86–1.1)	1.23
Thrombocytopenia (150–400 × 10^3^)	71 × 10^3^/mm^3^
Alanine aminotransferase (ALT) (0–34 UI/L)	194 UI/L
Aspartate aminotransferase (AST) (11–34 UI/L)	188 UI/L
Lactate dehydrogenase (LDH) (125–220 UI/L)	898 UI/L

## Data Availability

All relevant data have been presented in this manuscript, and further inquiry can be directed to the corresponding author.

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
