# Peer review of "Bilateral Serous Retinal Detachment as a Complication of HELLP Syndrome"

_diagnostics, 2023, doi:10.3390/diagnostics13091548_

Round 1

Reviewer 1 Report

This is a case report about a macular serous detachment related to hellp syndrome. This is quite interesting and nice. I would like you to comment which is the paper of corticosteroids, and could we only improve arterial pressure and the return to non-pregnant state could only lead to improvement of this syndrome. 

Everything is ok

Author Response

This is a case report about a macular serous detachment related to hellp syndrome. This is quite interesting and nice. I would like you to comment which is the paper of corticosteroids, and could we only improve arterial pressure and the return to non-pregnant state could only lead to improvement of this syndrome. 

Response: Thank you for your appreciations and recommendations. The text was corrected as recommended.

Reviewer 2 Report

In this paper, Teodoru et al. describe a case of HELLP syndrome with bilateral exudative retinopathy.

1. Hypertensive choroidopathy is well-described, yet never mentioned as a pathophysiological cause in relation to HELLP. The authors need to discuss hypertensive choroidopathy.

2. At several locations, the authors state "HELP" which should be "HELLP"

Dear Editor,

Thanks for the invitation to provide comments to this case report.

Not a novel case per se, as this is likely caused by hypertensive choriodopathy. But the relation to HELLP provide some novelty. I have few comments to the authors, of which #1 is major and requires extensive discussion.

Best wishes,

Yousif 

Author Response

In this paper, Teodoru et al. describe a case of HELLP syndrome with bilateral exudative retinopathy.

  1. Hypertensive choroidopathy is well-described, yet never mentioned as a pathophysiological cause in relation to HELLP. The authors need to discuss hypertensive choroidopathy.

Response: 

Thank you for your appreciations and recommendations.

The text has been improved in several sections, according to the recommendations.

  1. At several locations, the authors state "HELP" which should be "HELLP"

Response: The text was corrected as recommended.